# Exploring Viral–Host Protein Interactions as Antiviral Therapies: A Computational Perspective

**DOI:** 10.3390/microorganisms12030630

**Published:** 2024-03-21

**Authors:** Sobia Idrees, Hao Chen, Nisha Panth, Keshav Raj Paudel, Philip M. Hansbro

**Affiliations:** Centre for Inflammation, Centenary Institute and University of Technology Sydney, Faculty of Science, School of Life Sciences, Sydney, NSW 2007, Australia; sobia.idrees@uts.edu.au (S.I.); hao.chen@uts.edu.au (H.C.); n.panth@centenary.org.au (N.P.)

**Keywords:** protein–protein interactions, viral–host interactions, antiviral therapies

## Abstract

The interactions between human and viral proteins are pivotal in viral infection and host immune responses. These interactions traverse different stages of the viral life cycle, encompassing initial entry into host cells, replication, and the eventual deployment of immune evasion strategies. As viruses exploit host cellular machinery for their replication and survival, targeting key protein–protein interactions offer a strategic approach for developing antiviral drugs. This review discusses how viruses interact with host proteins to develop viral–host interactions. In addition, we also highlight valuable resources that aid in identifying new interactions, incorporating high-throughput methods, and computational approaches, ultimately helping to understand how these tools can be effectively utilized to study viral–host interactions.

## 1. Introduction

Protein–protein interactions (PPIs) play a crucial role in mediating various functions, including catalysing metabolic reactions, facilitating the transport of molecules, modifying the kinetic properties of enzymes, and altering the specificity of proteins [1,2]. In signalling events, proteins interact to regulate pathways and maintain essential cellular processes, contributing to cell growth and overall cellular function [3,4]. Over the years, numerous studies have been conducted to explore and predict comprehensive maps of PPIs in different organisms [5,6,7,8]. These interactions are fundamental to understanding the complexities of living organisms. The human body, for instance, relies on a vast network of proteins working in concert to maintain homeostasis, respond to stimuli, and carry out essential physiological functions. Disruptions in PPIs are often associated with diseases such as cancer, neurodegenerative disorders, viral infections, and immune system dysfunction [9,10]. PPIs are pivotal in determining the spatial and temporal organization of cellular processes. For example, the binding of signalling proteins can initiate cascades that regulate cellular responses to environmental cues. Enzymatic activities often involve the collaboration of multiple proteins, leading to metabolic pathways. Additionally, protein complexes play critical roles in cellular structures, such as the ribosome, which is essential for protein synthesis [11,12,13]. The majority of known PPIs involve domain–domain interactions (DDIs) mediated by globular domains in different proteins [14]. However, PPI detection experiments generate interaction data with the potential for false positives and false negatives. Exploring DDIs, where a domain in one protein interacts with a domain in another protein, can address these limitations and offer valuable insights into the complexities of biological systems [15]. The identification of DDIs relies on the three-dimensional structures of protein complexes available in the Protein Data Bank [16,17]. Databases like iPfam [18] and 3DID [19] contain extracted DDIs from known 3D structures. However, a limitation arises from the insufficient number of known 3D structures of proteins, prompting the development of computational methods for predicting DDIs. Despite the availability of various prediction methods, there is no unified platform for integrating predicted DDIs. In recent years, databases such as DOMINE [20] and UniDomInt [21] have emerged to specifically store DDIs from diverse resources. These databases provide a confidence score, enhancing the reliability of predicted DDIs. However, despite offering significant DDIs, these databases are outdated and lack recently published datasets. Another mode of interaction involves domain–motif interactions (DMIs), mediated by Short Linear Motifs (SLiMs) [22,23,24,25]. DMIs represent a subset of PPIs, where a domain of one protein interacts with a SLiM in another protein [26,27,28,29]. DMIs are often transient and play roles in various signalling processes, including protein targeting and signal transduction [30]. Specific SLiMs interact with specific domains to establish a DMI; for instance, proline-rich motifs tend to interact with SH3 domains [31]. SLiMs typically have 2–5 conserved positions essential for interaction with partner domains, offering flexibility in sequence patterns and enabling the establishment of different DMIs. This flexibility allows a single motif to bind to several domains from the same family, or variants of the same motif to bind with the same domain. For instance, PDZ domains interact with variants of the same motif (class I, class II, and class III), showcasing the promiscuity of DMIs, coupled with specificity that is often dependent on the sequence context of the motif, serving as a scaffold for establishing DMIs, while contextual residues define interaction specificity [32]. The promiscuous nature of DMIs is accompanied by a distinctive binding specificity. This specificity typically relies on the sequence context of the motif, acting as a scaffold for the establishment of DMIs, with contextual residues playing a crucial role in defining the specificity of interactions [33,34,35,36,37]. Pathogens, encompassing bacteria, viruses, protozoa, fungi, and helminths, interact with hosts, causing infections and resulting in disease development. Once inside the host cell, pathogens confront a robust immune defence system that works to confine and eliminate them [38]. At the viral–host interface, specific interactions between viral ligands and host cell receptors activate signalling cascades, leading to the recruitment of molecules crucial for immune responses. While these signalling pathways are essential for pathogen removal and minimal host damage, certain pathogens exploit them to manipulate the host immune response for their survival [39]. Targeting protein interactions has emerged as a critical strategy in the development of antiviral drugs. Disrupting key PPIs can impede viral replication, entry, and assembly, offering a promising avenue for designing effective therapeutics [38,40]. This review provides an overview of how viruses interact with host cell receptors, exploring how these interactions influence subsequent events that either support or eliminate the virus. Moreover, this review also discusses the current state of computational approaches and high-throughput methods that can help in identifying viral–host interactions. Furthermore, we discuss the importance of targeting specific protein interactions as a promising avenue for the development of antiviral drugs.

## 2. Fundamentals of Viral Protein Interactions

A comprehensive overview of viral proteins participating in key stages of the viral life cycle can help understand the molecular events critical for successful viral replication and propagation. During every stage, specialized viral proteins coordinate crucial functions, such as viral entry facilitated by attachment proteins and genome replication guided by polymerase enzymes and helicases. The assembly and packaging of viral particles involve structural proteins, while proteases play a pivotal role in the maturation of viral components. During egress, proteins mediate the release of newly formed virions from the host cell [41,42,43]. PPIs play a pivotal role in viral replication and assembly, guiding the proper packaging of genetic material and structural components [43,44,45]. A complete understanding of these interactions can not only shed light on the molecular mechanisms governing viral replication, but also provide potential targets for antiviral interventions aimed through disrupting essential PPIs [43]. While a considerable number of protein interactions involve globular domains and short linear peptide motifs (DMIs), targeting these interactions with small molecules has historically been challenging, leading to limited success. However, recent studies have identified potent inhibitors, including Obatoclax, ABT-199, AEG-40826, and SAH-p53-8, some of which are likely to receive approval as drugs. These inhibitors belong to diverse molecule classes, ranging from small molecules to peptidomimetics and biologicals [46]. In under two decades, three lethal coronaviruses—SARS-CoV, MERS-CoV, and SARS-CoV-2—have surfaced, resulting in a significant toll of hundreds of thousands of deaths. Additionally, other coronaviruses pose a threat to both domestic and wild animals through epizootics. With the longest genome among RNA viruses, members of this viral family express up to 29 proteins, engaging in intricate interactions with the host proteome. Understanding these interactions is crucial for identifying the cellular pathways exploited by these viruses to replicate and evade innate immunity [47,48,49,50]. A recent study focused on the relationship between host–virus PPIs, particularly on the involvement of disordered protein regions binding to folded protein domains in the virus life cycle. By employing proteomic peptide phage display, researchers have identified 281 peptides from intrinsically disordered regions of the human proteome that bind to eleven folded domains of SARS-CoV-2 proteins. Affinities for 31 interactions involving eight SARS-CoV-2 protein domains have been determined, with established key specificity residues for six interactions. Notably, two peptides inhibiting viral replication have been discovered, targeting Nsp9 and Nsp16. These findings highlight the potential of high-throughput peptide binding screens in simultaneously revealing host–virus interactions and identifying peptides with antiviral properties. Additionally, the prevalence of low-affinity interactions suggests that overexpressing viral proteins during infection may disrupt multiple cellular pathways [51]. Similarly, in one study, researchers identified six host targets, including CARD9 and CYP51A1, associated with both fungal infections and SARS-CoV-2 interactions, suggesting them as potential antiviral therapeutics [52]. In another study, researchers focused on the interaction between the Zika virus (ZIKV) and host cell machinery, particularly the ZIKV E protein domain III responsible for receptor binding. Through a yeast-2-hybrid screen, 21 proteins interacting with this domain have been identified, including the endoplasmic reticulum (ER) resident chaperone protein GRP78. The interaction was confirmed through co-immunoprecipitation and reciprocal co-immunoprecipitation, with immunofluorescence staining revealing co-localization between GRP78 and ZIKV E. Antibodies against GRP78’s N-terminus inhibited ZIKV entry, leading to reduced infection and viral production. A down-regulation of GRP78 by siRNA yielded similar results. This study suggested that GRP78 mediates ZIKV binding, internalization, and replication, and its up-regulation activates the unfolded protein response. Additionally, increased CHOP expressions and activations of caspases 7 and 9 were observed in response to ZIKV infection. These findings proposed the interaction between GRP78 and ZIKV E as a crucial factor in ZIKV infection and replication, potentially serving as a therapeutic target [53].

## 3. Characterizing Viral Host Interactions Using High-Throughput and Computational Approaches

Viral–host interactions can be characterized using three major experimental methods: yeast two-hybrid assays, co-immunoprecipitation, and mass spectrometry [54,55,56,57]. Yeast two-hybrid assays facilitate high-throughput screening and mimic in vivo conditions, making them valuable for detecting weak or transient interactions. However, they come with drawbacks like potential false positives and limitations in studying membrane–protein interactions [58]. Co-immunoprecipitation validates in vivo interactions under physiological conditions and is suitable for studying protein complexes, yet it may yield non-specific results and struggle with weak interactions [2]. Mass spectrometry, offering high sensitivity and quantitative data, excels in detecting protein complexes but requires sophisticated instrumentation [59,60]. Computational methods such as molecular dynamics simulations and network analysis complement experimental findings, providing a holistic view of interactions and aiding in the construction of refined models [61]. This integrated strategy is vital for advancing our understanding of viral pathogenesis and may guide the development of targeted antiviral interventions. The classical approach to drug discovery, focusing on small ligands interacting with well-defined binding sites in proteins like enzymes, ion channels, and receptors, has historically overlooked PPIs due to the inherent challenges in modulating them with small molecules [62,63]. Despite these challenges, targeting PPIs is increasingly considered a promising strategy for drug development. Some PPI modulators have entered clinical studies, with a few receiving marketing approvals, indicating the growing potential of targeting PPIs in drug discovery [64,65].

The accurate prediction of PPIs relies heavily on the integration of various data sources, each providing unique insights into the complex network of molecular associations within cells [40]. Structural data involve the three-dimensional arrangement of proteins, providing critical information about their physical interactions. Techniques such as X-ray crystallography and nuclear magnetic resonance (NMR) spectroscopy enable the determination of protein structures at an atomic resolution [66]. Structural data contribute valuable information about binding sites, interface residues, and the overall conformation of interacting proteins, offering a foundational understanding of the molecular basis of PPIs. Sequence data, encompassing amino acid sequences of proteins, are fundamental for predicting potential interactions. Sequence similarity and motif analysis aid in identifying conserved domains and regions crucial for binding interactions. Bioinformatic tools leverage sequence databases and algorithms to compare and align protein sequences, offering insights into the evolutionary relationships and conserved features that may dictate interactions (Table 1) [67]. Functional annotations provide context to protein interactions by associating biological functions and pathways. Gene ontology annotations, for example, categorize proteins based on their molecular functions, biological processes, and cellular components. Incorporating functional annotations can enhance the interpretation of predicted interactions, linking them to specific cellular processes and pathways [68]. Incorporating machine learning can also address these limitations, as machine learning algorithms can analyse large-scale datasets, identify patterns, and predict interactions with computational efficiency. These algorithms consider multiple features simultaneously, including sequence information, structural data, and functional annotations, aiming to enhance our understanding of the interactome and accelerate the identification of potential drug targets [69]. High-throughput experimental techniques generate large-scale datasets using systematically screening interactions. Methods such as yeast two-hybrid assays and mass spectrometry enable the identification of protein pairs engaging in physical interactions within a cellular context [70,71]. These experimental datasets serve as valuable training and validation sets for machine learning models, contributing to the refinement of PPI prediction algorithms. A holistic approach to PPI prediction involves the integration of diverse data types. Combining structural, sequence, and functional data with high-throughput experimental results can enhance the accuracy and reliability of predictions [72]. Therefore, to attain a complete understanding, the integration of experimental and computational approaches is crucial.

## 4. Targeting Viral–Host Protein Interactions for Therapeutics

Viruses strategically exploit cellular machinery through establishing virus–host protein interactions, essential for key stages such as entry (involving surface receptors and trafficking factors), genome replication and translation (facilitated by polymerases and translation factors), and egress (involving assembly and trafficking factors) [38]. Targeting and disrupting these interactions have emerged as promising avenues for therapeutic development against viral infections, as illustrated in Figure 1. This review encompasses various antiviral strategies, as outlined in Table 2. Remarkably, a significant proportion of approved antiviral drugs falls into the category of small molecule inhibitors. Their relatively compact size facilitates efficient cell penetration and precise interaction with specific binding sites. The practicality of oral administration further highlights their therapeutic viability. The journey toward developing small molecule inhibitors commences with the identification of viral proteins crucial to the virus life cycle. Central to this process is the acquisition of structural information essential for designing drugs specifically tailored to target viral proteins. Approaches for identifying protein structure in antiviral drug design include computer-aided virtual screening and experiment-based high-throughput screening. Computational drug discovery methods, such as virtual screening techniques involving cryo-electron microscopy, X-ray crystallography, and homology modelling, yield diverse protein structures. This aids molecular docking for the rapid identification of hit or lead compounds through screening databases like ZINC [84] and DrugBank [85]. However, it is essential to note the increasing prevalence of reported in silico docking studies and other computer-based predictions of antiviral activity without corresponding biological assays [86]. Nevertheless, it is strongly recommended to conduct biological validation to corroborate theoretical findings obtained through computational approaches. High-throughput screening involves the experiment-based identification of active small molecules within compound libraries, encompassing approved drugs, clinical trial candidates, and in-house databases. Despite being time-consuming and costly, this approach allows for the screening of a substantial number of compounds against a specific target viral protein, although success is not guaranteed. Following the identification of the target protein’s structure, initial hits or potential inhibitors undergo optimization to enhance their potency, selectivity, and pharmacokinetic properties. This optimization process involves modifying the chemical structure of hits to improve drug-like properties, such as adjusting functional groups, the molecular weight, and solubility. After medicinal chemistry optimization, lead compounds undergo an evaluation for absorption, distribution, metabolism, excretion, and toxicity (ADME/T) properties to ensure suitability for further development. Subsequently, in vivo/in vitro testing follows, culminating in the potential entry of promising candidates into clinical trials.

The development of antiviral drugs encounters numerous challenges, with only 106 drugs currently licensed for the therapy of viral diseases, despite the existence of over 200 human viruses. Approval for antivirals is limited to a handful of viruses, including HIV, HCV, influenza virus, RSV, HBV, HPV, herpesviruses, and SARS-CoV-2 [87]. Preclinical evaluations on animal models are crucial but often fraught with challenges, as many drugs that show promise initially fail to demonstrate efficacy in vivo or prove to be toxic to animals. Additionally, a considerable number of candidates do not pass the clinical trial phase. The use of in vitro systems that do not accurately represent the in vivo environment may contribute to such failures. Established cell lines and cell culture-adapted viral strains often differ significantly from normal primary cells and natural host cells. Novel in vitro tests that better mimic the in vivo environment, such as three-dimensional cultures, and the development of more accurate animal models are imperative for addressing these challenges in preclinical drug testing.

The one-drug-to-one-target paradigm has long been the cornerstone of drug development, focusing on finding drugs that inhibit specific targets. However, the numerous failures in this model, coupled with the extensive time (at least 12 years) and financial resources (around USD 3 billion per new drug approved) invested, highlight the need for a paradigm shift. The recent disappointment in clinical trials attempting to repurpose the drug combination lopinavir/ritonavir for SARS-CoV-2 treatment exemplifies the challenges associated with designing highly specific monotargeted drugs [88,89]. The pan-antiviral strategy refers to a novel therapeutic approach that targets multiple viruses, acting as a broad-spectrum inhibitor rather than focusing on a specific viral species. This strategy diverges from the traditional one-drug-to-one-target paradigm, which often faces high failure rates and substantial financial costs. Pan-antivirals, by design, aim to disrupt various pathways essential for the survival and infectivity of different viruses [88]. Identifying pan-antivirals is possible through efficient computational models, specifically 2D ligand-based approaches. Perturbation theory and machine learning (PTML) models, recognized for anti-HIV predictions, stand out for drug repurposing. Additionally, alignment-free multitarget (AFMT) models efficiently screen large chemical libraries for potential pan-antivirals [88]. Thus, the pan-antiviral strategy holds the potential to revolutionize antiviral therapy, providing a more effective and efficient strategy against a broad spectrum of viral infections.

**Table 2 microorganisms-12-00630-t002:** Overview of therapeutic strategies for disrupting viral–host protein interactions.

Strategy	Sub-Strategy	Mechanism	Example
Small molecule inhibitor	Protease inhibitor	Preventing the maturation of viral proteins by targeting viral proteases involved in the cleavage of viral polyproteins	Ritonavir and lopinavir for HIV infection [90]
	Entry inhibitor	Preventing viral entry by blocking the interaction between viral envelope proteins and host cell receptors	Maraviroc for HIV infection [91]
	Helicase/unwinding inhibitor	Inhibiting viral replication by targeting viral helicases involved in unwinding viral RNA	Remdesivir for SARS-CoV-2 infection [92]
Antibody and immunotherapy	Monoclonal antibody	Blocking interaction with host cells by specifically binding to viral proteins	Palivizumab for RSV infection [93]
	Convalescent plasma therapy	Extracting neutralizing antibodies against the virus from plasma from recovered individuals	Convalescent plasma for Ebola virus infection [94]
Peptide-based inhibitor	Peptide mimetic	Inhibiting competitively with the host cell proteins using peptides that mimic key regions of viral proteins	Enfuvirtide for HIV infection [95]
	Fusion inhibitor	Preventing the fusion of viral and host cellular membranes by targeting viral fusion proteins	
RNA-based therapy	RNA interference	Degrading viral RNA or interfering with viral RNA translation by introducing small interfering RNA or short hairpin RNA	TKM-130803 for Ebola infection [96]
	Antisense oligonucleotide	Preventing protein synthesis using a synthetic oligonucleotide that binds to viral RNA	Fomivirsen for cytomegalovirus retinitis [97]
Host cell factor targeting	Kinase inhibitor	Targeting host cell kinases involved in viral replication	Baricitinib for SARS-CoV-2 infection [98]
	Host cell receptor modulation	Modifying host cell receptors to decrease viral binding and entry	Maraviroc for HIV infection [91]
Viral protein degradation	Proteolysis targeting chimera	Designing molecules that induce the degradation of specific viral proteins by recruiting cellular degradation machinery	Telaprevir for hepatitis C [99]
Vaccine	Subunit vaccine	Developing vaccines based on specific viral proteins to induce an immune response against those proteins	HPV prophylactic vaccines [100]
	mRNA vaccine	Using mRNA to instruct host cells to produce viral antigens, hence stimulating an immune response	mRNA vaccines for SARS-CoV-2 infection [101]
CRISPR/Cas9-based therapy		Editing viral genomes to disrupt essential viral protein interactions through CRISPR/Cas9 approach	In early development stage
Repurposed drug		Identifying existing drugs with antiviral properties that can disrupt viral protein interactions	Remdesivir initially developed for Ebola, later repurposed for SARS-CoV-2 infection [102]

## 5. Challenges and Considerations in Targeting Viral Protein Interactions

Protein interactions play a central role in various cellular processes and are crucial for the functioning of biological systems. Targeting these interactions holds great promise for the development of therapeutic interventions. However, several challenges and considerations must be addressed to translate research findings into clinically effective treatments [38]. PPIs are considered potential drug targets, although converting small molecules into therapeutics poses challenges, requiring a delicate balance to induce therapeutic effects without adverse consequences. The degradation of TP53 by MDM2, prevented by Nutlin-3, exemplifies this delicate balance in PPIs. A multidisciplinary approach, combining genetics, proteomics, biochemical, and biophysical methods to understand the role of PPIs in viral diseases can accelerate drug discovery [103]. Numerous computational techniques have emerged to predict PPIs; however, most of these approaches were designed for predicting interactions within the same species rather than across different species. Methods tailored for intra-species PPI prediction lack the ability to differentiate between interactions among proteins within the same species and those involving proteins from different species. Consequently, they are not suitable for anticipating inter-species PPIs [104]. One of the significant challenges lies in bridging the gap between laboratory discoveries and clinical applications. Many potential drug candidates identified through research may face obstacles in transitioning to effective treatments due to complexities in replicating experimental conditions in clinical settings. The conventional approach to small-molecule drug discovery primarily centres on targeting protein–ligand interactions involving enzymes, ion channels, or receptors due to their well-defined ligand-binding sites. However, modulating PPIs through small molecules has historically been challenging, often labelling PPIs as “undruggable” targets. The human interactome is estimated to encompass 130,000–650,000 types of PPIs, outnumbering enzymes, and receptors. Designing small molecules for PPI interfaces presents challenges such as through the larger hydrophobic interface area (1500–3000 Å2), flat and groove-poor topology, continuous or discontinuous amino acid residues, and the absence of endogenous ligands for reference. Additionally, drugs targeting PPIs have a higher molecular weight (>400 Da), posing difficulties in meeting established criteria like Lipinski’s “rule of 5” compared to traditional small-molecule drugs (200–500 Da). These factors collectively contribute to the complexity of developing small molecules that effectively modulate PPIs [9,65]. The complex nature of protein interactions within living organisms adds complexity to therapeutic development. The in vivo environment may introduce unforeseen variables, making it challenging to predict how interventions targeting specific interactions will behave within the human body [3]. Efforts to manipulate protein interactions can inadvertently affect other biological processes, leading to off-target effects. This concern emphasizes the importance of specificity in drug design to minimize unintended consequences [4]. Ensuring the safety of targeted protein interactions is a critical consideration. Comprehensive safety assessments are essential for identifying and mitigating potential adverse effects on normal cellular functions, organ systems, or overall physiological homeostasis [5]. Advancements in computational biology play a vital role in predicting potential off-target effects and optimizing drug design. Virtual screening and molecular dynamics simulations aid in assessing the selectivity and safety profiles of candidate compounds [6]. Utilizing relevant biological assays and model systems that closely mimic human physiology can enhance the predictive value of preclinical studies. This approach facilitates a more accurate assessment of the potential clinical impact of interventions targeting specific protein interactions [7]. Considering the multifaceted nature of viral infections, combination therapies targeting multiple protein interactions or pathways may provide enhanced efficacy while minimizing the risk of resistance development [8]. Addressing these challenges and considerations is imperative for the successful development of antiviral therapeutics targeting protein interactions [9].

## 6. Conclusions

The exploration of viral-host interactions as a foundation for antiviral therapies can serve as a rich landscape of opportunities for targeted intervention. This review provides an overview of viral–host protein interactions and highlights the importance of a comprehensive approach to drug development. It also suggests that a combination of high-throughput methods and computational approaches can facilitate the identification of novel interactions, which can in turn provide a deeper understanding of the viral protein network.

## Figures and Tables

**Figure 1 microorganisms-12-00630-f001:**
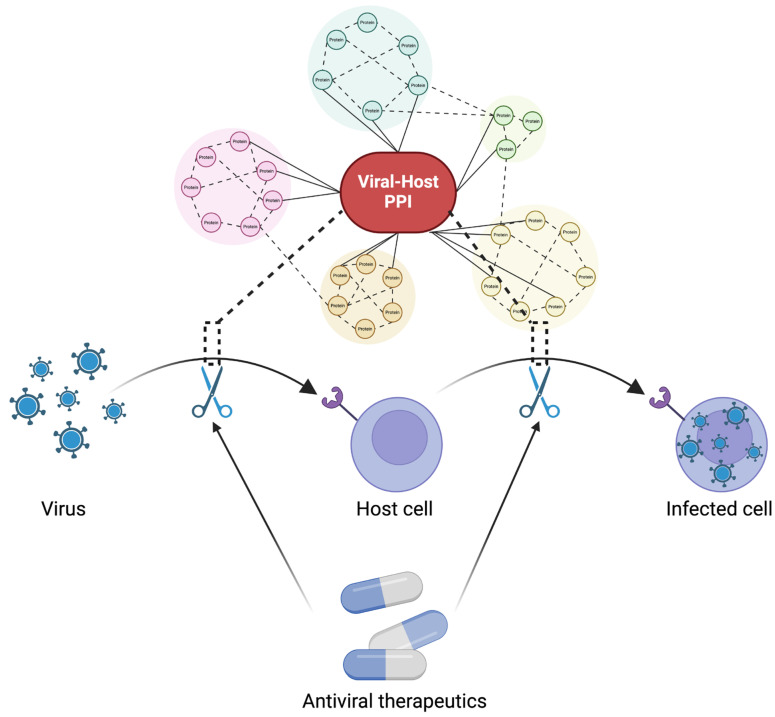
Schematic representation illustrating the potential points of action for antiviral interventions targeting protein–protein interactions (PPIs). Antivirals can exert their effects during two distinct phases of the viral life cycle: (1) at the initial interaction between viruses and surface markers, when viruses are attempting to enter host cells, and (2) post entry, when viruses have successfully entered host cells.

**Table 1 microorganisms-12-00630-t001:** Resources/repositories used to study viral–host interactions.

Resource/Repository	Description	URL
VirusMentha [73]	Integrated resource for viral–host and protein–protein interactions.	https://virusmentha.uniroma2.it/ [Accessed on 10 January 2024]
VirHostNet3.0 [74]	Network-based tool for exploring virus–host interactions. Integrates interaction, annotation, and pathway data.	https://virhostnet.prabi.fr/ [Accessed on 10 January 2024]
HPIDB3.0 [75]	Human Protein Interaction Database with information on virus–host interactions.	https://hpidb.igbb.msstate.edu/ [Accessed on 10 January 2024]
VirBase v3.0 [76]	Database focusing on interactions between viral and host miRNAs, proteins, and genes.	https://www.rna-society.org/virbase/ [Accessed on 10 January 2024]
Virus-Host DB [77]	The Virus-Host DB systematically structures information pertaining to the associations between viruses and their hosts, presenting it in the format of paired NCBI taxonomy IDs for both viruses and their respective hosts.	https://www.genome.jp/virushostdb [Accessed on 10 January 2024]
VHRdb [78]	The Viral Host Range database (VHRdb) provides experimentally validated interactions.	https://viralhostrangedb.pasteur.cloud/ [Accessed on 10 January 2024]
AIMaP [79]	AIMaP is a database and web server for users to easily explore an atlas of interactions between SARS-CoV-2 macromolecules and hosts.	https://mvip.whu.edu.cn/aimap/home/ [Accessed on 10 January 2024]
VirusMint [80]	A repository containing information on interactions between proteins of viruses and humans.	https://maayanlab.cloud/Harmonizome/resource/Virus+MINT [Accessed on 10 January 2024]
PHISTO [81]	Platform for studying infection mechanisms through Pathogen–Host Interactions (PHIs)	https://phisto.org/ [Accessed on 10 January 2024]
HVIDB [82]	HVIDB is a computational platform predicting human-virus protein interactions based on multiple data sources.	http://zzdlab.com/hvidb/ [Accessed on 10 January 2024]
HVPPI [72]	HVPPI offers a thorough annotation of human-virus protein interactions along with online tools for functional PPI analysis.	http://bio-bigdata.hrbmu.edu.cn/HVPPI/ [Accessed on 10 January 2024]
Viruses.STRING [83]	Viruses.STRING is a database for virus–virus and virus–host protein interactions, combining experimental and text-mining data for probability assessments.	viruses.string-db.org [Accessed on 10 January 2024]

## Data Availability

Not applicable for review article.

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
