# Peer review of "Exploring Viral–Host Protein Interactions as Antiviral Therapies: A Computational Perspective"

_microorganisms, 2024, doi:10.3390/microorganisms12030630_

Round 1
Reviewer 1 Report
Comments and Suggestions for Authors
The authors focus on important host-pathogen interactions, a topic of great interest to a wide scientific audience. However, upon closer examination, it looks like the manuscript predominantly focuses on computational methodologies, which does not fulfill the expectation of its title, 'Exploring host-pathogen interactions as antiviral therapies.' To enhance clarity and specificity, the title needs to be more specific with emphasis on the computational aspects.
Moreover, the manuscript predominantly focuses on viral infections, lacking significant coverage of other pathogens. The provided resources, although potentially valuable, lack the description of practical application through real-world examples. Additional note: the manuscript seems to lack the comprehensive coverage of the previous literature. For example: https://www.ncbi.nlm.nih.gov/pmc/articles/PMC9364728/. It seems there is more literature that can be included. This highlights the necessity for a more detailed literature search for this manuscript.
Regrettably, the readability of the article suffers from an excess of verbosity, making it very vague. Was ChatGTP used? The authors use long sentences that lack clarity in their message. In other words, the authors utilize big words and long sentences but do not say much. For example, the conclusion of the article has long sentences without telling specifically what this review describes and why this review is important.
In conclusion, I recommend a thorough revision focusing on simplicity and precision in language, accompanied by clear and concise statements. By adopting a more straightforward approach and providing clear summaries of previous literate and available resources, the manuscript can significantly enhance its accessibility and impact.
Comments on the Quality of English LanguageThe readability of the article suffers from an excess of verbosity, making it very vague. Was ChatGTP used? The authors use long sentences that lack clarity in their message. In other words, the authors utilize big words and long sentences but do not say much. For example, the conclusion of the article has long sentences without telling specifically what this review describes and why this review is important.
Author Response
The authors thank the editor, the editorial team, and the reviewers for considering our manuscript and for their valuable suggestions. We genuinely appreciate their guidance in improving the quality of our work and have incorporated their suggestions to enhance the rigour and clarity of our manuscript. We have now addressed all the comments and incorporated the changes into a revised version (highlighted in red font) as described in the point-by-point responses below.
Reviewer 1:
The authors focus on important host-pathogen interactions, a topic of great interest to a wide scientific audience. However, upon closer examination, it looks like the manuscript predominantly focuses on computational methodologies, which does not fulfill the expectation of its title, 'Exploring host-pathogen interactions as antiviral therapies.' To enhance clarity and specificity, the title needs to be more specific with emphasis on the computational aspects.
Author response: We agree with the reviewer. We have now changed the title of the paper for clarity.
Moreover, the manuscript predominantly focuses on viral infections, lacking significant coverage of other pathogens. The provided resources, although potentially valuable, lack the description of practical application through real-world examples. Additional note: the manuscript seems to lack the comprehensive coverage of the previous literature. For example: https://www.ncbi.nlm.nih.gov/pmc/articles/PMC9364728/. It seems there is more literature that can be included. This highlights the necessity for a more detailed literature search for this manuscript.
Author response: We agree with the reviewer. We have changed the title and focus of the review to discuss viral-host protein interactions. We have also incorporated the latest literature related to viral-host interactions.
Regrettably, the readability of the article suffers from an excess of verbosity, making it very vague. Was ChatGTP used? The authors use long sentences that lack clarity in their message. In other words, the authors utilize big words and long sentences but do not say much. For example, the conclusion of the article has long sentences without telling specifically what this review describes and why this review is important.
Author response: We agree with the reviewer. There were issues with the clarity of sentences. We have now thoroughly revised the text for better clarity. We did not use ChatGPT.
In conclusion, I recommend a thorough revision focusing on simplicity and precision in language, accompanied by clear and concise statements. By adopting a more straightforward approach and providing clear summaries of previous literate and available resources, the manuscript can significantly enhance its accessibility and impact.
Author response: We have now revised the manuscript and conclusion section for clarity.

Reviewer 2 Report
Comments and Suggestions for Authors
Idrees et al. explored and highlighted the importance of host-pathogen interaction-based protein networks in developing antiviral drugs using several computational networks. If this review includes the identification of pan-viral therapy based on a protein network- the idea is to find common deregulated proteins among a similar group of viruses and if it could lead to the design and testing of new drugs.
Author Response
The authors thank the editor, the editorial team, and the reviewers for considering our manuscript and for their valuable suggestions. We genuinely appreciate their guidance in improving the quality of our work and have incorporated their suggestions to enhance the rigour and clarity of our manuscript. We have now addressed all the comments and incorporated the changes into a revised version (highlighted in red font) as described in the point-by-point responses below.
Reviewer 2 Idrees et al. explored and highlighted the importance of host-pathogen interaction-based protein networks in developing antiviral drugs using several computational networks. If this review includes the identification of pan-viral therapy based on a protein network- the idea is to find common deregulated proteins among a similar group of viruses and if it could lead to the design and testing of new drugs.
Author response: We have now discussed pan-antiviral therapy and why its important. Location: Page 7.
